# Is It Safe to Treat Stable Patients with Bacteremic Urinary Tract Infections with High-Resistant-Rate Antibiotics?

**DOI:** 10.3390/diagnostics14151620

**Published:** 2024-07-26

**Authors:** Zvi Shimoni, Hanna Salama, Talya Finn, Paul Froom

**Affiliations:** 1The Adelson School of Medicine, Ariel University, Ariel 4070001, Israel; zshimoni@laniado.org.il (Z.S.); tfinnfried@laniado.org.il (T.F.); 2Sanz Medical Center, Laniado Hospital, Netanya 4244916, Israel; 3Internal Medicine Department A, Sanz Medical Center, Laniado Hospital, Netanya 4244916, Israel; hanasalama3008@gmail.com; 4Clinical Utility Department, Sanz Medical Center, Laniado Hospital, Netanya 4244916, Israel; 5School of Public Health, University of Tel Aviv, Tel Aviv 6997801, Israel

**Keywords:** bacteremic urinary tract infection, ceftriaxone, internal medicine, bacterial resistance, safety, length of stay

## Abstract

Background and Objectives: In most areas of the world, urine bacteria have high resistance rates to third-generation cephalosporins, and it is unclear if it is safe to treat stable patients with bacteremic urinary tract infections (UTI) with those antibiotics. There are recommendations that empiric therapy for a suspected UTI should include only antibiotics with resistance rates less than 10%. Materials and Methods: In this historical observational single center study, we selected 180 stable internal medicine patients hospitalized between January 2019 and December 2021, with identical bacteria isolated from blood and urine cultures. Charts were reviewed to determine if deaths and readmissions up to 30 days after discharge were due to bacterial resistance to initial antibiotic therapy (BRIAT). Results: The patient’s median age was 82 years (1st–3rd quartiles, 73–87 years). A total of 54.4% were female. There were 125 patients treated with ceftriaxone. A total of 38 (30.3%) had BRIAT. Four patients died, but none were because of a delay in appropriate treatment. The median days of hospitalization for all patients was 7 days, and 9 days versus 6 days in those with and without BRIAT. There were no re-hospitalizations for a UTI in patients with BRIAT. Conclusions: We conclude that, despite high resistance rates, empiric ceftriaxone in stable hospitalized patients with a bacteremic UTI is safe. There was no urosepsis-related mortality during the hospitalization or on follow-up. The treatment of all patients with wider-spectrum antibiotics might have decreased the median hospital stay by only one day. The potential effect would be even lower if all patients with a suspected systemic UTI were treated with wide-spectrum antibiotics, because some patients do not have an infection of the urinary tract. A reassessment of the recommendation that empiric therapy for a suspected systemic urinary tract infection should include only wider-spectrum antibiotics is warranted.

## 1. Introduction

The hospitalization rate of febrile elderly patients can be as high as 80%, and the best treatment option is unclear in those without an extra-urinary tract source [1,2,3]. There are claims that empiric treatment for a suspected systemic urinary tract infection (UTI) should include only antibiotics with resistance rates of <10% [4,5,6], precluding the use of third-generation cephalosporins in most areas of the world [7]. However, despite high resistance rates, many hospitals use empiric therapy with cephalosporins [8,9,10,11]. It is essential to show that bacterial resistance to initial antibiotic therapy (BRIAT) does not reduce survival or treatment success among stable, non-ICU patients [12], which would support the use of agents with high resistance rates and lower the use of empiric wide-spectrum antibiotics. 

In our hospital, most of the patients with a suspected UTI are treated with ceftriaxone despite resistance rates around 30% for *Escherichia coli*, and 40–50% for *Klebsiella pneumonia* and *Proteus mirabilis* [9], which in previous studies did not cause in-hospital deaths in patients with and without bacteremia despite BRIAT [13,14]. However, others have shown an increased mortality rate [8], and further studies are warranted to ensure that in-hospital deaths are not related to empiric ceftriaxone treatment in the face of bacterial resistance [8,12].

In the following study, we selected consecutive patients with a bacteremic urinary tract infection, including patients with low blood pressure, to determine if BRIAT increases the risk for in-hospital mortality and 30-day readmissions after discharge.

## 2. Materials and Methods

In this historical prospective observational study, we selected consecutive patients, hospitalized in one of three internal medicine departments at Laniado Hospital from 1 January 2019 to 31 December 2021, that had identical bacteria isolated from blood and urine cultures and no other cause for their hospitalization. Patients are treated empirically with antibiotics upon admission to the emergency department and changes are later made if necessary (based on culture results). The standard treatment recommendation is 3–7 days with intravenous antibiotics that cover the bacteria grown in culture, and 7–14 days with antibiotics including the recommended oral antibiotics on discharge. We excluded all patients admitted with hypotension who did not respond to fluids, were treated with vasopressors or were mechanically ventilated. The following variables were extracted from the electronic database and the charts: age, gender, referral from a nursing home, a history of urinary tract infections, significant urological pathology (benign prostatic hypertrophy, a permanent urinary catheter, bladder cancer, prostatic cancer, renal cancer, urethral stricture, or a nephrostome), and fever in the hospital (≥38 °C or <36 °C). Presenting symptoms were classified according to those consistent (yes/no) with a urinary tract infection (dysuria, hematuria, abdominal or costal vertebral pain, urgency, and difficulty urinating). We also extracted blood pressure, and hypotension was a systolic blood pressure <100 mm Hg. Other admission laboratory tests were extracted and binned according to the normal reference ranges and clinically significant cut-off values [15].

Urine cultures were processed using standard microbiologic methods, and isolates were identified by the VITEK 2 system (bioMerieux, Marcy l’Etoile, France). A positive culture required the presence of at least 10^5^ colony-forming units per milliliter of urine. Resistance to ceftriaxone was characterized by a minimal inhibitory concentration (MIC) of ≥4 mg/mL or ESBL positivity.

The outcome variables were in-hospital deaths, length of stay, and readmission within 30 days of discharge. All charts were reviewed to determine if a delay in appropriate antibiotic therapy contributed to the in-hospital deaths and readmissions.

### Statistical Analysis

We determined rates, means with standard deviations, medians with 1st and 3rd quartiles, and odds ratios with 95% confidence limits. We divided the patients into those treated with ceftriaxone and those treated with wider-spectrum antibiotics. To determine selection bias, we compared the independent variables in those treated with ceftriaxone to those treated upfront with other antibiotics. Logistic regression determined the odds that BRIAT in patients treated with ceftriaxone increased extended hospitalizations, before and after adjustment for other risk factors. All independent variables not adding significantly to the model were removed and added back one at a time, and then retained if they added significantly to the model.

## 3. Results

There were 196 patients with the same organism in their blood and urine. We excluded 13 on mechanical ventilators or with septic shock, 2 patients admitted because of a cerebral vascular accident, and 1 post-operative patient, leaving a cohort of 180 patients. The patients were elderly, often not alert, frequently came from a nursing home and had a history of a UTI (Table 1). The median length of stay was 7 days.

Over 70% of the patients had a nonspecific presentation without urinary tract symptoms (Table 2) and 82.2% presented with fever. The most common non-specific complaints were fever and general deterioration, but 6.7% presented with syncope/falls. There were six patients whose chief complaint was fever, but did not have fever in the hospital. Therefore, there were 16.1% (29/180) patients who did not have either fever or urinary tract symptoms.

Comparing patients with and without urinary tract symptoms, only the frequency of low serum albumin and referral from a nursing home were significantly different (Table 3). Those with urinary tract symptoms had a trend for being more alert and being febrile, as well as those with a permanent catheter and with a urological diagnosis. Upon logistic regression, only the nursing home referrals were negatively associated with the presence of urinary tract symptoms. None of the other variables added back one at a time significantly added to the model.

Of the infections, 70.6% were due to E coli and 34.6% were ESBL positive, whereas for all the infections, 37.8% were ESBL positive (Table 4). Leukocyte esterase was positive in 94.4% (170/180), and nitrite in 56.1% (101/180). There were 1.6% (3/180) of patients with a negative dipstick urinalysis. Physicians requested a urine culture in the three febrile patients despite the negative dipstick, and the negative test result did not delay antibiotic therapy.

There were 125 patients treated empirically with ceftriaxone. The others were treated with broader spectrum antibiotics (a carbapenem (*n* = 14), amikacin (*n* = 2), gentamicin (*n* = 10), and piperacillin/tazobactam (*n* = 25)) given because of previous culture results (*n* = 17), low blood pressure (*n* = 7), uropathology (*n* = 4), neutropenic fever (*n* = 1), and for no apparent reason in 20 patients. A total of 4 patients received chloramphenicol as part of end-of-life treatment. BRIAT occurred in 14.2% (2/14) of patients treated with a carbapenem, in 0% (0/10) treated with gentamicin, in 36.0% (9/25) treated with piperacillin/tazobactam, in 50% (1/2) treated with amikacin, and in all patients treated with chloramphenicol.

Antibiotics were changed because of BRIAT in 28.8% of those receiving ceftriaxone and in 29.1% of those who were treated with other antibiotics (Table 5). Culture results allowed for de-escalation in 12.8% of those receiving primarily wider-spectrum antibiotics, i.e., the wider-spectrum antibiotics were given unnecessarily. Only 2 patients had antibiotics changed because of a lack of response, and 1 because of an allergic reaction.

There were 66 patients with uropathology, including a permanent catheter (*n* = 26), benign prostate hypertrophy (*n* = 36), cancer (*n* = 15), or a nephrostome (*n* = 3). Compared to those treated with other antibiotics, patients treated with ceftriaxone had similar ages, rates of referral from nursing homes, and hypotension on admission. However, they had significantly less uropathology, more urological-related symptoms, fewer patients with a previous UTI, fewer patients with BRIAT (Table 6), and more patients with anemia (Table 7). There were no significant differences in white blood counts, C-reactive proteins, lactate dehydrogenase, serum creatinine, blood urea nitrogen, and plasma glucose concentrations.

Patients treated with ceftriaxone who had BRIAT had an odds of 6.93 (Table 6) of a prolonged hospitalization. On multivariate analysis, the odds ratios (95% confidence intervals) for hospitalization of 10 days or more for those treated with ceftriaxone were 8.22 (2.99–22.62) for BRIAT after adjustment for uropathology (3.45 (1.18–10.04)) and low serum albumin concentrations (1.81 (0.98–3.34)). The median value for days of hospitalization was 7 days (95% confidence interval 5 to 10 days), 9 (7–14) and 6 (5–7) in those with and without BRIAT. Therefore, the potential decrease in hospitalization days was one, if all patients were treated with a wide-spectrum antibiotic.

Four patients treated initially with ceftriaxone died, three deaths occurred in patients with BRIAT treated initially with ceftriaxone, changed to piperacillin/tazobactam in two patients and ertapenem in the other according to culture results. They did not die of a UTI because of a delay in “appropriate” antibiotic therapy (Table 8). One was an 81-year-old female with dementia (case 1) with urinary retention, who responded to antibiotics and died suddenly 19 days after admission. Another patient (case 2), a 63-year-old cachectic bedridden female patient who also responded to antibiotics, died from other causes 11 days after admission. Lastly, a 99-year-old female (case 3) responded to antibiotics but died of aspiration pneumonia. The fourth patient was a 94-year-old male who had an infection sensitive to ceftriaxone but died after one day.

Two patients treated with other antibiotics on admission died. A 69-year-old female did not have BRIAT but died of urosepsis (case 5), and an 88-year-old male admitted for end-of-life care had BRIAT (case 6) but died of other causes. Of the patients excluded, 12/17 died, so for all patients with a bacteremic urinary tract infection, the death rate was 9.6% (19/197).

There were 35 patients treated with ceftriaxone who presented with hypotension. Antibiotics were changed because of BRIAT in 10, and none died because of a delay in “appropriate” antibiotics.

The prediction of ESBL-producing organisms was poor. There were 38.8% (68/180) bacteria with ESBL positivity. Upon univariate analysis, only a previous UTI, anemia, hypoalbuminemia, and an elevated BUN were significantly associated with ESBL positivity (Table 9). There was a positive trend for a referral from a nursing home, a previous hospitalization, uropathology, and serum creatinine ≥ 2 mg/dL. The other variables were not associated with ESBL positivity (not shown in the table; being alert, febrile, a systolic blood pressure ≤100 mmHg, and the other laboratory tests).

Only a previous urinary tract infection and a hemoglobin <10 gm/dL significantly added to the logistic regression model, with increased odds for predicting ESBL positivity of 2.77 (1.44–5.31) and 3.69 (1.74–7.89), respectively. ESBL positivity was 27.9% (19/90) in those without either anemia or a previous urinary tract infection, 59.1% (13/22) in those with anemia, 48.0% (24/50) in those with a previous UTI, and 66.7% (12/18) in patients with both anemia and a previous urinary tract infection.

There were no readmissions for a UTI in patients with BRIAT over the 30-day follow-up period.

## 4. Discussion

The major finding of this study was that hospitalized patients with a bacteremic UTI treated empirically with ceftriaxone did not die of urosepsis. This is despite a high resistance rate, and including patients with hypotension on admission and other abnormal laboratory results. In patients with BRIAT, the length of hospital stay increased by a median of three days, with an 8-fold risk of a prolonged hospitalization (10 days or more) after adjustment for the other risk factors (uropathology and low serum albumin).

The strength of this study is that it included chart views of all patients, including those who died, and that there was a 30-day re-hospitalization follow-up period. Another strength of the study is that we only included patients with bacteremic UTIs. Other studies of patients with a systemic UTI include patients with bacteriuria without a clear extra-urinary source, an undetermined number of whom will not have a UTI. In the elderly, a positive urine culture is not an abnormal finding, because reported frequencies in healthy elderly men and women have been reported to be 6% and 16.5%, respectively [16]. One study reported that 29.6% of febrile hospitalized geriatric patients with a clear extra-urinary source had a positive urine culture [17].

Other studies of UTIs that included patients with significant comorbidities reported discordant associations with BRIAT and short- or longer-term mortality. A previous study reviewed the charts of 934 elderly patients with a UTI, excluding patients with hypotension; 222 had bacteremia; 2/3 of the 316 responded to therapy, and no patient died from urosepsis [14]. Other studies have also found no increase in the mortality of BRIAT in those hospitalized with a urinary tract infection [12,18,19,20,21,22]. Babich et al. [21] found that BRIAT was not associated with a 30-day mortality of 30.8% in 315 patients with a catheter-associated UTI. Wiggers et al. [22] studied 469 patients with a bacteremic UTI; 21.5% had inappropriate antibiotic therapy that was not associated with the 4.3% mortality rate at 7 days or a 9.6% rate at 30 days. Eliakim-Raz et al. [12] reported on 981 young and old patients in a multicenter study where inappropriate antibiotic therapy was given in 47% of patients with a complicated UTI, not associated with the 8.7% mortality rate by day 30. In that study, mortality was related to septic shock, intensive care unit admission, corticosteroid treatment, being bedridden, older age metastatic cancer and a catheter-associated UTI, but not with inappropriate antibiotic therapy. The authors of those studies concluded that because there was no observable benefit of early appropriate empirical treatment on survival rates, physicians might consider limiting the use of antibiotics generally held in reserve if the patient is stable.

On the other hand, in Spain, 29.3% of 270 patients 75 years old or older with severe UTIs received inappropriate antibiotics, and the in-hospital death rate was 8.9% [23] Inappropriate antibiotics were associated with a 3.5 increase in odds for in-hospital deaths after adjustment for Apache II ≥15, dementia and neoplasia. Mark et al. found an association between BRIAT and an increase in 30-day mortality [8]. Others have reported an increase in mortality in patients with BRIAT bacteremia, including patients with a urinary source [24]. However, controlling for co-morbidities in statistical models does not prove an association between BRIAT and mortality. There were no chart reviews in those studies to determine if BRIAT caused hospital deaths. This is important because most hospital deaths in those with a systemic UTI, as in our study, are unrelated to the infection [25]. To show an association with BRIAT, a chart review needs to confirm that the death was due to septic shock or end-organ damage that occurred because of the delay in “appropriate” antibiotic treatment. We are unaware of such reports.

Although BRIAT did not result in in-hospital deaths, it is unclear if initial empirical treatment with ceftriaxone or other narrower spectrum antibiotics rather than broader spectrum antibiotics with <10% resistance rates (only carbapenems or amikacin in our hospital [9]) is warranted, because of the increase in the length of hospital stay. If a carbapenem was substituted, we would expect a 3-day lower length of stay in those with BRIAT [8,13,14,18], as was found in this study. Most patients did not have BRIAT, and if all patients in this study were treated at admission with wider-spectrum antibiotics, the median length of hospitalization would have decreased by only one day. The potential decrease in the days of hospitalization for all patients with a suspected systemic UTI would be even less, because not all febrile patients without evidence of disease outside the urinary tract have a UTI. Shimoni et al. reported that 40% of hospitalized patients with fever from an unknown source had a negative urine culture, many of whom did not require antibiotic therapy [17]. A watch-and-wait policy could lower the days of hospitalization in some patients [26]. Therefore, although wider-spectrum antibiotics would benefit a subgroup of patients, they would have a negligible overall effect on the length of hospitalization. A reassessment of the recommendation that empiric therapy for a suspected systemic urinary tract infection includes only antibiotics with resistance rates < 10% is warranted.

We found that around 70% of the patients did not have urinary tract symptoms. The most common non-specific complaints were fever and general deterioration, but patients also presented with syncope/falls. This is consistent with other studies of patients with a bacteremic UTI [27,28,29,30,31], including those who are mentally stable [32]. We found that nonspecific symptoms were associated with referrals from a nursing home and hypoalbuminemia, but patient characteristics cannot be used to determine when to suspect a UTI. This emphasizes the problem of underdiagnosing a UTI in elderly patients hospitalized with nonspecific complaints that can negatively affect the correct diagnosis [31].

One way to identify patients with ESBL-producing organisms and improve empiric antibiotic use is to identify risk factors. We found that only anemia and a previous UTI predicted ESBL positivity, with risks of 29.9% to 66.7% for those without and with both those risk factors, and in a study of patients with and without a bacteremic UTI we reported that risk factors for BRIAT, mostly due to ESBL positivity, were male sex, history of a UTI, a referral from a nursing home and the use of a permanent urethral catheter, with risk values from 13.0% to 68.9% as the number of the risk factors increased [14]. Despite the extensive literature attempting to predict ESBL positivity, Timbrook TT et al. [33] concluded in a systematic review that ESBL derivation studies have a median sensitivity and specificity of 55% and 94%, with external validations reflecting a lower median sensitivity and specificity of 32% and 90%, respectively, and that the two ESBL models with multiple external validations performed poorly, with c-statistics of 71% or less. No one has shown the clinical utility of using the risk factors to change therapy. It appears that using clinical indicators is not sensitive or specific enough to change antibiotic therapy, and the ability to rapidly identify the subgroup with ESBL-producing organisms awaits rapid genetic and other identification systems that are sensitive, specific [34] and affordable.

The major limitation of this study is that patients with uropathology and lower albumin tests were less likely to receive ceftriaxone. The markers of severity (laboratory test results), however, were not significantly different in those selected to receive broader-spectrum antibiotics. They still had around a 30% rate of BRIAT that did not cause in-hospital deaths. It appears, therefore, that the selection treatment bias did not affect the results, and that it would be safe to treat all the stable patients with a bacteremic UTI with ceftriaxone.

Secondly, it is also unclear if our findings can be extrapolated to other settings. Detailed patient characteristics, including presentations, however, allow others to consider if the results apply to their patients. Thirdly, after the culture results, treatment for an additional 7 days was given according to bacteria sensitivities. Only two patients had antibiotics changed because they did not respond to initial therapy, and perhaps those who responded despite BRIAT did not need “appropriate” antibiotics for an additional 7 days. One study demonstrated that 5–7 days of antibiotics is equivalent to longer durations [35] for patients with pyelonephritis. Fourthly, we cannot assume that the results can be extrapolated to other cephalosporins.

Another limitation is that the follow-up was in a single hospital without contact with the patients or their families. We did not have readmission data from other hospitals, but 88% of readmissions in Israel were shown to be to the same regional hospital [36]. No patient was re-hospitalized with a UTI after BRIAT, nor discharged with significant end-organ damage due to sepsis. Finally, there were only eight patients treated empirically with ceftriaxone with the combination of hypotension and BRIAT. It may be prudent to treat those patients empirically with a carbapenem. Much larger cohorts of patients are warranted to demonstrate that patients with hypotension do not require treatment empirically with a carbapenem.

## 5. Conclusions

We conclude that stable hospitalized patients with a bacteremic UTI treated empirically with ceftriaxone, despite high resistance rates, did not die because of urosepsis during the hospitalization, including patients with hypotension on admission. The delay in concordant antibiotic treatment with bacterial sensitivities increased the risk of a prolonged hospitalization by 8-fold, but not the risk of readmissions. If all patients, however, were treated with wider-spectrum antibiotics, there would only be a one-day potential decrease in hospitalization days, which would be even less if all patients with a suspected UTI were included. A reassessment of the recommendation that empiric therapy for a suspected systemic urinary tract infection includes only antibiotics with resistance rates < 10% is warranted.

## Figures and Tables

**Table 1 diagnostics-14-01620-t001:** Patient characteristics.

Variable	*N* = 180 Patients
Median (1st–3rd quartiles)	
Age	82 (73–87)
Length of stay	7 (5–10)
	Number (%)
Female gender	98 (54.4)
Permanent catheter	29 (10.3)
Not alert	91 (50.6)
Nursing home	48 (26.7)
History of UTI	70 (38.9)

**Table 2 diagnostics-14-01620-t002:** Presenting symptoms.

Symptoms	Total *N* (%)	Fever *N* (%)
Dysuria	28 (15.6)	19 (67.9)
Hematuria	5 (2.8)	4 (80.0)
Abdominal pain	12 (6.7)	9 (75.0)
Difficulty urinating	11 (6.1)	11 (100)
Urgency	1 (0.6)	1 (100)
Total urinary tract symptoms *	57 (31.7)	44 (77.2)
Fever only	46 (25.5)	40 (87.0)
Syncope/falls	12 (6.7)	5 (41.7)
Nausea/vomiting	8 (4.4)	6 (75.0)
Shortness of breath	8 (4.4)	4 (50.0)
Hypotension	4 (2.2)	2 (50.0)
General deterioration	42 (23.3)	34 (81.0)
Transient ischemic attack	3 (1.7)	3 (100)
Total	180	148 (82.2)

* one of the above.

**Table 3 diagnostics-14-01620-t003:** Patients with urinary tract symptoms compared to the other patients.

Variables *	None *N* = 123	Yes *N* = 57	Odds Ratio (95% CI)
SBP < 100 mmHg	37 (30.1)	10 (17.5)	0.49 (0.23–1.08)
Age 80 years or more	73 (59.3)	34 (59.6)	1.01 (0.54–1.92)
Females	69 (56.1)	29 (50.9)	0.81 (0.43–1.52)
Alert	58 (47.2)	31 (54.4)	1.34 (0.71–2.51)
Previous UTI	45 (36.6)	23 (40.4)	1.12 (0.74–1.70)
Fever	49 (86.0)	99 (80.5)	1.48 (0.62–3.54)
Hemoglobin <10 gm/dL	31 (25.2)	9 (15.8)	0.56 (0.25–1.26)
Albumin <3.5 g/dL	55 (44.7)	13 (22.8)	0.37 (0.18–0.75)
Creatinine ≥2 mg/dL	34 (27.6)	9 (15.8)	0.49 (0.22–1.11)
Permanent catheter	15 (12.2)	9 (15.8)	1.35 (0.55–3.30)
Urological diagnosis	41 (33.3)	25 (43.9)	1.56 (0.82–2.97)
Nursing home	39 (31.7)	9 (15.8)	0.40 (0.18–0.90)

* SBP—systolic blood pressure, UTI—urinary tract infection.

**Table 4 diagnostics-14-01620-t004:** Bacteria and extended-spectrum beta-lactamases.

Bacteria	Total *N* (%)	ESBL *N* (%)
*Escherichia coli*	127 (70.6)	44 (34.6)
*Klebsiella species*	21 (11.7)	13 (61.9)
*Proteus species*	16 (8.9)	5 (31.3)
*Pseudomonas species*	5 (2.8)	2 (40.0)
*Citrobacter species*	3 (1.7)	1 (33.3)
*Providencia species*	3 (1.7)	2 (66.7)
*Staphylococcus aureus*	3 (1.7)	0 (0.0)
*Enterococcus faecalis*	1 (0.6)	1 (100)
*Enterobacter cloaca*	1 (0.6)	0 (0.0)
Total	180	68 (37.8)

**Table 5 diagnostics-14-01620-t005:** Reason antibiotics were changed.

Antibiotics	Ceftriaxone *N* = 125 *n* (%)	Other *N* = 55 *n* (%)	Total *N* = 180 *n* (%)
No change	83 (66.4)	23 (52.7)	106 (58.9)
BRIAT *	36 (28.8)	16 (29.1)	52 (28.9)
De-escalation	0 (0.0)	13 (23.6)	13 (12.8)
No response	2 (1.6)	0 (0.0)	2 (1.1)
Unclear	3 (2.4)	3 (5.5)	6 (3.3)
Allergy	1 (0.8)	0 (0.0)	1 (0.6)

* Bacterial resistance to initial antibiotic therapy.

**Table 6 diagnostics-14-01620-t006:** Ceftriaxone and treatment with other antibiotics.

Variables **	Ceftriaxone *N* = 125	Other *N* = 55	Odds Ratio (95% CI) *
Age ≥ 80 years	78 (62.4)	29 (52.7)	1.49 (0.78–2.82)
Female	74 (59.2)	24 (43.6)	1.87 (0.99–3.56)
Previous UTI	37 (29.6)	31 (56.4)	0.33 (0.17–0.63)
Uropathology	35 (28.0)	31 (56.4)	0.30 (0.16–0.58)
Fever	105 (84.0)	43 (78.2)	1.47 (0.66–3.26)
UT symptoms	47 (37.6)	10 (18.2)	2.71 (1.25–5.88)
Previous UTI	37 (29.6)	31 (56.4)	0.33 (0.17–0.63)
Nursing home	33 (26.4)	15 (27.3)	0.96 (0.47–1.95)
Hypotension	32 (25.6)	15 (27.3)	0.92 (0.45–1.88)
BRIAT	38 (30.4)	16 (29.1)	1.06 (0.53–2.13)
ESBL	35 (28.0)	33 (60.0)	0.26 (0.13–0.50)
Death	3 (3.2)	1 (5.5)	0.88 (0.58–1.34)
LOS * ≥ 10 days	28 (22.4)	17 (30.9)	0.65 (0.32–1.31)
BRIAT-yes	18/38 (47.4)	7/16 (43.8)	
BRIAT-no	10/87 (11.5)	10/39 (25.6)	
Odds ratios	6.9 (2.8–17.3)	2.3 (0.7–7.6)	

* 95% CI—95% confidence interval. ** UTI—urinary tract infection, UT—urinary tract, BRIAT—bacterial resistance to initial antibiotic therapy, ESBL—extended spectrum beta-lactamase, LOS—length of stay.

**Table 7 diagnostics-14-01620-t007:** Ceftriaxone versus other antibiotics; laboratory tests.

Laboratory Tests *		Ceftriaxone *n* = 125	Other *n* = 55	Odds Ratio (95% CI) *
WBC (10^9^ cells/L)	<12 12–14.9 ≥15	57 (46.6) 30 (24.0) 38 (30.4)	28 (50.9) 10 (18.2) 17 (30.9)	1.07 (0.74–1.54)
HB (gm/dL)	≥12 10–11.9 <10	62 (49.6) 41 (32.8) 22 (17.6)	20 (36.4) 17 (30.9) 18 (32.7)	1.64 (0.43–0.95)
Platelets (10^9^/L)	<100	6 (4.8)	3 (5.5)	0.87 (0.21–3.64)
CRP (mg/dL)	<10 10–99 100–199 ≥200	5 (4.0) 50 (40.0) 38 (30.4) 32 (25.6)	4 (7.3) 23 (41.8) 17 (30.9) 11 (20.0)	1.20 (0.83–1.73)
LDH (U/L)	<400 400–599 ≥600	54 (43.2) 50 (40.0) 21 (16.8)	23 (41.8) 24 (43.6) 8 (14.5)	1.02 (0.65–1.58)
Albumin (gm/dL)	≥3.5 3.0–3.49 <3.0	84 (67.2) 24 (19.2) 17 (13.6)	28 (50.9) 16 (29.1) 11 (20.0)	0.68 (0.45–1.02)
Bilirubin (mg/dL)	≥1.2	19 (15.2)	3 (5.5)	3.11 (0.88–10.97)
Creatinine (mg/dL)	≥2.0	28 (22.4)	15 (27.3)	0.77 (0.37–1.59)
BUN (mg/dL)	<20 20–29 ≥30	26 (20.8) 37 (29.6) 62 (49.6)	11 (20.0) 16 (29.1) 28 (50.9)	0.97 (0.65–1.45)
Glucose (mg/dL)	<150 150–199 ≥200	78 (62.4) 30 (24.0) 17 (13.6)	40 (72.7) 9 (16.4) 6 (10.9)	1.31 (0.82–2.10)

* WBC—white blood cells, HB—hemoglobin, CRP—c-reactive protein, LDH—lactic dehydrogenase, BUN—blood urea nitrogen, CI = confidence interval.

**Table 8 diagnostics-14-01620-t008:** In-hospital deaths.

Case	Age/Sex *	Organism	Changed to	Days
Ceftriaxone initial treatment				
1	81F	E-coli-ESBL	Piperacillin/ tazobactam	19
2	63F	E coli-ESBL	Piperacillin/ tazobactam	11
3	99F	Citrobacter-ESBL	Ertapenem	7
4	94M	Staph aureus ****	none	1
	Other antibiotics			
5	69F	E coli-ESBL **		7
6	88M	E coli ***	Ceftriaxone	5

* F = female, M = male. ** treated initially with piperacillin/tazobactam. *** treated initially with chloramphenicol. **** sensitive to ceftriaxone.

**Table 9 diagnostics-14-01620-t009:** Associated variables with ESBL positivity.

Variables	ESBL + *N* = 68	Other *N* = 112	Odds Ratio
Age ≥ 80 years	39 (57.4)	68 (60.7)	0.87 (0.47–1.60)
Female	36 (52.9)	62 (55.4)	0.91 (0.50–1.66)
Nursing home	20 (29.4)	28 (25.0)	1.25 (0.64–2.45)
Hospitalization <90 days before	13 (19.1)	12 (10.7)	1.97 (0.84–4.61)
Uropathology	28 (41.2)	38 (33.9)	1.36 (0.73–2.54)
Urethral catheter	12 (17.6)	12 (10.7)	1.79 (0.75–4.24)
Previous UTI	36 (52.9)	32 (28.6)	2.81 (1.50–5.27)
Hemoglobin <10 gm/dL	25 (36.8)	15 (13.4)	3.76 (1.81–7.83)
Albumin <3 gm/dL	17 (25.0)	11 (9.8)	3.06 (1.33–7.02)
BUN ≥30 mg/dL	43 (63.2)	47 (42.0)	2.38 (1.28–4.42)
Creatinine ≥ 2 mg/dL	20 (29.4)	23 (20.5)	1.61 (0.81–3.23)

## Data Availability

The data presented in this study are available on request from the corresponding author.

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
