# Peer review of "Is It Safe to Treat Stable Patients with Bacteremic Urinary Tract Infections with High-Resistant-Rate Antibiotics?"

_diagnostics, 2024, doi:10.3390/diagnostics14151620_

Round 1
Reviewer 1 Report
Comments and Suggestions for Authors
Please find below my comments on the manuscript:
-The length of the abstract exceeds the word limit of the Diagnostics journal which is 200 words. Please rearrange based on journal instructions.
-Mortality was all-cause mortality or infection-related? Please specify.
-The type of study should be explained better, was it observational/interventional? Insert also information such as single-center, clinical study.
-Table 2, insert abbreviations and their full form in the table footnote.
Author Response
Please find below my comments on the manuscript:
-The length of the abstract exceeds the word limit of the Diagnostics journal which is 200 words. Please rearrange based on journal instructions.
Done
-Mortality was all-cause mortality or infection-related? Please specify.
line 83 "All charts were reviewed to determine if a delay in appropriate antibiotic therapy contributed to the in-hospital deaths and readmissions."
-The type of study should be explained better, was it observational/interventional? Insert also information such as single-centre, clinical study.
We added "In this historical, observational, single center study"
-Table 2, insert abbreviations and their full form in the table footnote.
Done
We hope our manuscript is now acceptable for publication,
Thanks,
Reviewer 2 Report
Comments and Suggestions for Authors
When ceftriaxone (CTRX) is used as initial treatment for urinary tract infection (UTI), even if the bacteria are resistant to CTRX, it does not worsen the prognosis, only prolongs the hospital stay. Based on these results, you disagree with the opinion that treatment should be started with antibiotics that are less resistant to UTI. This is very interesting.
There are some unclear points in the description of the results, and the paper would be better if those were cured.
Major Revision
Page 3, Line 112- and Table 2: It would be easier to understand if you clarify what urinary tract symptoms refer to. In Table 2, from “Dysuria” to “Urgency” are consider to be urinary tract symptoms, but they are grouped together with “Total urinary tract symptom”, which makes it difficult to understand.
Page 4, Line 127- : In the text, the ESBL rate is 37.8%, but in Table 3 it is 34.6%. Is the denominator different?
Page5, Line 138- and Table 4: The rate of BRIAT other than CTRX is written, but it would be easier to understand if the rate for CTRX is also written. Also, in Table 4, the number of BRIAT patient from other antibiotics is 10 cases, but form the text it can be read that there are 12+4(chloramphenicol).
Page 7, Line 172- : You write 3 deaths, but there are 4 cases in Table 4. Please provide details about Case 4 as well.
Minor Revision
Page 4, Line 122 : without urinary tract only the frequency → without urinary tract symptom only the frequency ?
Page4, Line123- : You describe as “UTI symptom”, but up until then you describe as “urinary tract symptom”
There are two “Table 2”. Therefore, the subsequent Table numbering is all out of alignment. Please correct the numbers and change the text regarding them.
Page 9, Line256: There is no ")".
Author Response
Reviewer 1.
When ceftriaxone (CTRX) is used as initial treatment for urinary tract infection (UTI), even if the bacteria are resistant to CTRX, it does not worsen the prognosis, only prolongs the hospital stay. Based on these results, you disagree with the opinion that treatment should be started with antibiotics that are less resistant to UTI. This is very interesting.
Thank you,
There are some unclear points in the description of the results, and the paper would be better if those were cured.
Major Revision Page 3, Line 112- and Table 2: It would be easier to understand if you clarify what urinary tract symptoms refer to.
We added to line 73, in the materials and methods section, all presenting symptoms consistent with a UTI -(dysuria, hematuria, abdominal or costal vertebral pain, urgency, and difficulty urinating).
In Table 2, from “Dysuria” to “Urgency” are consider to be urinary tract symptoms, but they are grouped together with “Total urinary tract symptom”, which makes it difficult to understand.
I changed the table lines and added *one of the above at the end of the table. I hope it is now clear.
Page 4, Line 127- : In the text, the ESBL rate is 37.8%, but in Table 3 it is 34.6%. Is the denominator different?
We changed the text to "Of the infections 70.6% were due to E coli, 34.6% ESBL positive, whereas for all the infections, 37.8% were ESBL positive (Table 4)."
Page5, Line 138- and Table 4: The rate of BRIAT other than CTRX is written, but it would be easier to understand if the rate for CTRX is also written.
Perhaps the misunderstanding was due to the errors in the chart. The BRIAT rate for ceftriaxone is 36/125 or 28.8%, and for the others 16/55 of 29.1% is written in the table and the text now.
Also, in Table 4, the number of BRIAT patient from other antibiotics is 10 cases, but form the text it can be read that there are 12+4(chloramphenicol).
Thank you for finding our error! The table 4 was changed , it was actually 16 in total, and now consistent also with table 5
Page 7, Line 172- : You write 3 deaths, but there are 4 cases in Table 4. Please provide details about Case 4 as well.
The 4 cases are shown, but only 3 are BRIAT, and the fourth case did not have an antibiotic change because it was sensitive to ceftriaxone. I added that in the table for clarity ****sensitive to ceftriaxone. I also added the description in the text.
There were 4 patients who died treated initially with ceftriaxone, 3 deaths in patients with BRIAT treated initially with ceftriaxone and changed to piperacillin/tazobactam in two patients and ertapenem in the other according to culture results. They did not die of a UTI because of a delay in "appropriate" antibiotic therapy (Table 7) One was an 81-year-old demented female (case 1) with urinary retention who responded to antibiotics and died suddenly 19 days after admission. Another patient (case 2) a 63-year-old cachectic bedridden female patient who also responded to antibiotics died from other causes 11 days after admission. Lastly, a 99-year-old female (case 3) responded to antibiotics but died of aspiration pneumonia. The fourth patient was a 94-year-old male who had an infection sensitive to ceftriaxone, but died after one day.
Minor Revision Page 4, Line 122 : without urinary tract only the frequency → without urinary tract symptom only the frequency ?
Don't understand the comment, I think the sentence is ok after changing the UTI symptoms to urinary tract symptoms
Page4, Line123- : You describe as “UTI symptom”, but up until then you describe as “urinary tract symptom”
Corrected to urinary tract symptoms, thanks.
There are two “Table 2”. Therefore, the subsequent Table numbering is all out of alignment. Please correct the numbers and change the text regarding them.
Done.
Page 9, Line256: There is no ")"
Done
Round 2
Reviewer 2 Report
Comments and Suggestions for Authors
Thank you for taking my opinions into consideration and revising your paper. Your paper is now clearer, and I find your paper useful. Thank you for your hard work.